# COVID-19: Impact of Original, Gamma, Delta, and Omicron Variants of SARS-CoV-2 in Vaccinated and Unvaccinated Pregnant and Postpartum Women

**DOI:** 10.3390/vaccines10122172

**Published:** 2022-12-17

**Authors:** Fabiano Elisei Serra, Elias Ribeiro Rosa Junior, Patricia de Rossi, Rossana Pulcineli Vieira Francisco, Agatha Sacramento Rodrigues

**Affiliations:** 1Disciplina de Obstetrícia, Departamento de Obstetrícia e Ginecologia, Faculdade de Medicina FMUSP, Universidade de São Paulo, São Paulo 05508-070, Brazil; 2Faculdade de Medicina, Universidade de Santo Amaro (UNISA), São Paulo 04743-030, Brazil; 3Gerência de Obstetrícia, Hospital Maternidade Interlagos, São Paulo 04802-190, Brazil; 4Departamento de Estatística, Universidade Federal do Espírito Santo, 514 Fernando Ferrari Avenue, Vitória 29075-910, Brazil; 5Department of Perinatology and Gynecology, Mandaqui Hospital, São Paulo 02401-400, Brazil

**Keywords:** COVID-19, SARS-CoV-2, vaccines, pregnancy, postpartum period, puerperal period, obstetrics, obstetric population, variants of concern

## Abstract

This study compares the clinical characteristics and disease progression among vaccinated and unvaccinated pregnant and postpartum women who tested positive for different variants of severe acute respiratory syndrome coronavirus 2 (SARS-CoV-2) using the Brazilian epidemiological data. Data of pregnant or postpartum patients testing positive for SARS-CoV-2 and presenting with coronavirus disease 2019 (COVID-19) from February 2020 to July 2022 were extracted from Brazilian national database. The patients were grouped based on vaccination status and viral variant (original, Gamma, Delta, and Omicron variants), and their demographics, clinical characteristics, comorbidities, symptoms, and outcomes were compared retrospectively. Data of 10,003 pregnant and 2361 postpartum women were extracted from the database. For unvaccinated postpartum women, intensive care unit (ICU) admission was more likely; invasive ventilation need was more probable if they tested positive for the original, Gamma, and Omicron variants; and chances of death were higher when infected with the original and Gamma variants than when infected with other variants. Vaccinated patients had reduced adverse outcome probability, including ICU admission, invasive ventilation requirement, and death. Postpartum women showed worse outcomes, particularly when unvaccinated, than pregnant women. Hence, vaccination of pregnant and postpartum women should be given top priority.

## 1. Introduction

Coronavirus disease (COVID-19) is caused by severe acute respiratory syndrome coronavirus 2 (SARS-CoV-2) and was first reported in Wuhan, China, in December 2019. COVID-19 was declared a pandemic in March 2020 [1]. As of 14 October 2022, over 620 million cases and 6.5 million deaths have been reported worldwide [2].

Morbidity and mortality among pregnant women and perinatal outcomes are of critical concern. COVID-19 affects several systems, with a clinical spectrum ranging from asymptomatic to severe and critical diseases. Initial studies did not report a higher mortality rate or more severe disease in obstetric patients with COVID-19 compared with the general population; however, later studies reported that obstetric patients had a higher risk of intensive care admission, requiring invasive ventilation, and death than non-pregnant women [3,4,5,6,7,8,9,10,11].

During the course of the pandemic, SARS-CoV-2 developed mutations, resulting in the emergence of new variants. Variants of concern (VOCs) have higher transmissibility, increased virulence, and altered COVID-19 epidemiology and presentation. The effectiveness of public health, social measures, and available therapeutics decrease for VOCs [12]. As of 30 July 2022, three VOCs have been detected in Brazil: The Gamma, Delta, and Omicron variants [13].

A high number of maternal deaths due to COVID-19 has been reported in Brazil; 124 deaths as of 18 June 2020 [14].

In a previous study, we performed a retrospective analysis of the records of the System of Information about Epidemiological Surveillance of Influenza (SIVEP-Gripe), and, as of 2 January 2021, 295 maternal deaths were linked to COVID-19, which was confirmed using reverse transcriptase polymerase chain reaction (RT-PCR). The study included data of female patients aged 10 to 49 years hospitalized due to severe COVID-19 disease (confirmed positive using RT-PCR for SARS-CoV-2), from 17 February 2020 to 2 January 2021. All cases were divided into three groups: pregnant, puerperal, and neither pregnant nor puerperal. After general comparisons, adjustments for confounding variables (propensity score matching [PSM]) were made, using demographic and clinical characteristics, admission to intensive care unit (ICU) and invasive ventilatory support), and outcome (cure or death). After PSM, the prognosis in puerperal women was worse than that in pregnant women with respect to admission to ICU, invasive ventilatory support, and death, with OR (95% CI) 1.97 (1.55–2.50), 2.71 (1.78–4.13), and 2.51 (1.79–3.52), respectively. Outcomes in postpartum women were similar to those in non-pregnant and non-puerperal women [15].

The Brazilian Obstetric Observatory for COVID-19 was created on 7 April 2021; it provides updated data released by the Ministry of Health of Brazil. In total, 1031 maternal deaths due to COVID-19 (680 pregnant women and 351 postpartum women) were reported in Brazil as of 5 May 2021 [16]. When vaccination campaigns began for pregnant and postpartum women in Brazil on 1 May 2021, compared with non-maternal women and with men, maternal patients were at a significantly higher risk of suffering from severe symptoms of COVID-19, of requiring ICU admission and intubation, and of dying of COVID-19 in the first 17 epidemiological weeks of 2021 than in the first year of pandemic. [17,18].

Since puerperal women with COVID-19 had a worse prognosis than pregnant women, and vaccination is one of the most effective strategies to reduce the transmission and severity of infectious diseases, it is important to evaluate the effectiveness of the vaccines in the obstetric population and their protection against the new VOCs [19,20]. Therefore, this study aimed to compare the clinical characteristics and outcomes of pregnant and postpartum women based on vaccination status and VOC using data from a Brazilian national database.

## 2. Materials and Methods

Patient data were extracted from the System of Information about Epidemiological Surveillance of Influenza (SIVEP-Gripe), a Brazilian national database that includes surveillance data regarding severe acute respiratory syndrome (SARS) [21]. Compulsory reporting of SARS due to the flu (acute respiratory conditions characterized by at least two of the following signs and symptoms: fever, chills, headache, sore throat, cough, coryza, anosmia, and ageusia), associated with dyspnea, persistent chest pressure, oxygen saturation (SpO_2_) < 95% on room air, or cyanosis is included in this database. All public and private hospitals are obligated to report all hospitalizations and deaths due to SARS-CoV-2.

The SIVEP-Gripe includes demographic data (sex, age, skin color or ethnicity, level of education, obstetric status, and city of residence), clinical data (signs and symptoms, risk factors, and comorbidities), epidemiological data (previous flu vaccinations, community-acquired infection, or nosocomial infection), and laboratory and etiological diagnoses. Hospital and ICU admission, requirement for ventilatory support (invasive or non-invasive), and disease outcome (death or cure) are also reported. SIVEP-Gripe records are publicly available, anonymized data. Therefore, according to the Brazilian Ethics regulatory requirements, ethical approval by an Institutional Review Board was not necessary for this study. This study followed the principles of the Declaration of Helsinki.

In this study, the database was searched for records from 17 February 2020 to 16 July 2022 (epidemiological weeks 8–53 in 2020; 1–52 in 2021; and 1–28 in 2022), as the first Brazilian records began in epidemiological week 8 of 2020 (symptom onset for the first confirmed case of COVID-19 in Brazil occurred on 17 February 2020) [22]. The search included all records of obstetric patients aged 10 to 49 years who were hospitalized with COVID-19 (confirmed using RT-PCR for SARS-CoV-2) (Figure 1). A total of 12,364 women were divided into two groups: pregnant (*n* = 10,003) and puerperal (*n* = 2361). Each group was subdivided based on the dominant VOCs in Brazil at the time of diagnosis: original, Gamma, Delta, and Omicron. Dominance occurs when more than 50% of cases are related to a particular variant. The Gamma variant was considered dominant from 1 February 2021 to 31 July 2021, the Delta variant from 1 August 2021 to 31 December 2021, and the Omicron variant from 1 January 2022 to 16 July 2022 [13]. Vaccination of pregnant and postpartum patients began in May 2021 in Brazil, during the dominance of the Gamma variant [18]. Therefore, patients in the Gamma, Delta, and Omicron subgroups were also divided based on the vaccination status. Only valid responses of the analyzed variable were considered. The tables of analyses show the number of valid observations for each variable. Variables used in the analysis were age, skin color or ethnicity, level of education, comorbidities, signs and symptoms, ICU admissions, invasive respiratory support requirement, and outcome (death). The reported comorbidities included chronic cardiovascular disease, asthma, obesity, and diabetes. The reported signs and symptoms included fever, cough, sore throat, dyspnea, respiratory discomfort, SpO_2_ < 95% on room air, diarrhea, vomiting, abdominal pain, fatigue, anosmia, and ageusia.

Abbreviations: SIVEP-Gripe, System of Information about Epidemiological Surveillance of Influenza; SARS, severe acute respiratory syndrome; COVID-19, coronavirus disease 2019; RT-PCR, reverse transcriptase polymerase chain reaction.

### 2.1. Data Analysis

Categorical variables are presented as numbers (*n*) and percentages (%). These were compared using the chi-square or Fisher’s exact test. The odds ratio (OR) and 95% confidence interval (CI) were calculated to compare the relative odds of the occurrence of a variable of interest between the groups. All statistical analyses were conducted using R software (version 4.0.3; R Foundation for Statistical Computing, Vienna, Austria) [23].

### 2.2. Patient and Public Involvement

Public data were used in this study. No patient was recruited or involved in the research. Patients were not invited to contribute to the designing, writing, or editing process at any point throughout the study.

## 3. Results

Data from 12,364 obstetric women (age range: 10–49 years), who were hospitalized with COVID-19 that was confirmed using RT-PCR for SARS-CoV-2, were included in this study. The pregnant group included 10,003 patients, while the puerperal group included 2361 patients. Patient age and education level were not significantly different between the groups (Table 1). However, the rates of infection with the Omicron variant in different ethnic groups were significantly different between the pregnant and puerperal groups. Patients in the puerperal group had fewer comorbidities.

Among unvaccinated patients who tested positive for the original SARS-CoV-2 variant, the puerperal group, compared with the pregnant group, had a lower frequency of fever (OR: 0.76; 95% CI: 0.65–0.90), coughing (OR: 0.69; 95% CI: 0.58–0.83), vomiting (OR: 0.47; 95% CI: 0.33–0.65), and ageusia (OR: 0.72; 95% CI: 0.53–0.97), and a higher frequency of respiratory discomfort (OR: 1.19; 95% CI: 1.01–1.39), SpO_2_ < 95% on room air (OR: 1.70, 95% CI: 1.44–2.00), ICU admission (OR: 1.82; 95% CI 1.54–2.13), invasive respiratory support (OR: 2.62; 95% CI 2.13–3.22), and death (OR: 2.48; 95% CI: 1.96–3.13) (Table 2 and Table 3). Unvaccinated patients in the puerperal group infected with the Gamma variant had a lower frequency of fever (OR: 0.61; 95% CI: 0.50–0.76), coughing (OR: 0.56; 95% CI: 0.44–0.71), dyspnea (OR: 0.75; 95% CI: 0.60–0.94), vomiting (OR: 0.49; 95% CI: 0.32–0.73), anosmia (OR: 0.60; 95% CI 0.43–0.82), and ageusia (OR: 0.69; 95% CI: 0.50–0.94); however, they were more likely to be admitted to the ICU (OR: 1.58; 95% CI: 1.29–1.93), require invasive respiratory support (OR: 2.17; 95% CI: 1.75–2.70), and die (OR: 1.99; 95% CI: 1.56–2.53), compared with unvaccinated pregnant patients. There were no differences in symptoms, ICU admission, requirement for invasive respiratory support, or death between the two groups among unvaccinated patients with the Delta variant. Unvaccinated patients in the puerperal group with the Omicron variant had a lower frequency of vomiting (OR: 0.23; 95% CI: 0.03–0.98) and a higher frequency of ICU admission (OR: 2.61; 95% CI: 1.25–5.38) and invasive respiratory support (OR: 3.09; 95% CI: 1.12–8.50), compared with unvaccinated pregnant patients with the Omicron variant.

Among vaccinated patients with the Gamma variant, patients in the puerperal group were more likely to die than those in the pregnant group (OR: 2.58; 95% CI: 1.03–6.06). Among vaccinated patients with the Delta variant, symptoms, ICU admission, requirement for invasive respiratory support, and death rate were not significantly different between the pregnant and puerperal groups. Patients in the puerperal group with the Omicron variant who were vaccinated had a lower frequency of fever (OR: 0.43. 95% CI: 0.28–0.66) and vomiting (OR: 0.35; 95% CI: 0.12–0.85) and a higher frequency of dyspnea (OR: 1.64; 95% CI: 1.07–2.50), SpO_2_ < 95% on room air (OR: 1.97; 95% CI: 1.20–3.22), ICU admission (OR: 1.91; 95% CI: 1.13–3.17), and invasive respiratory support (OR: 2.88; 95% CI: 1.34–6.21), compared with those in the pregnant group.

## 4. Discussion

In this study, the clinical findings and outcomes of pregnant and postpartum women with COVID-19 differed. The predominant VOCs at the time of diagnosis (original, Gamma, Delta, and Omicron variants) and patient vaccination status influenced these differences.

### Postpartum Women

When the COVID-19 pandemic began in 2020, it was believed that pregnant and puerperal women were not at a higher risk of adverse outcomes or death than the non-obstetric population [3,4,24,25,26,27]. However, updated studies reported higher rates of invasive ventilation, ICU admissions, and death due to COVID-19 in the obstetric population [6,7,8,10,28,29]. Differences in clinical findings and outcomes between pregnant and postpartum women are unclear, as previous studies regarding these differences included non-hospitalized patients in whom a COVID-19 diagnosis was not confirmed using RT-PCR [8,9,10,11].

Another previous study compared the clinical findings and outcomes of pregnant, puerperal, and non-pregnant non puerperal women aged 10 to 49 years with COVID-19 from the beginning of the pandemic until 2 January 2021. This previous study used PSM and found that puerperal women were more likely than pregnant women to be admitted to the ICU (OR: 1.97), require invasive respiratory support (OR: 2.71), and die (OR: 2.51) [15].

The three delays model can be used to explain why postpartum women had worse prognoses than pregnant women in this study [30]. During the postpartum period, women may delay seeking medical assistance as they focus on taking care of their new-born, ignoring their own health [31,32]. Moreover, the risk of thromboembolism increases during the postpartum period as well as in patients with COVID-19 [33,34,35,36,37]. As the SIVEP-Gripe database does not provide information regarding the date of delivery or the gestational age upon diagnosis, some patients in the puerperal group may have been infected with SARS-CoV-2 during pregnancy that progressed to SARS after delivery. In addition, pregnant women with SARS may terminate their pregnancy as a therapeutic measure and may die during the puerperal period. Caesarean sections account for more than 57% of deliveries in Brazil and are even more frequent among patients with severe COVID-19, which may increase the risk of maternal mortality in this population [37,38,39,40].

Among the unvaccinated patients in this study, those in the puerperal group had a worse prognosis than those in the pregnant group for each VOC. In the puerperal group, 35.6%, 48.0%, 51.0%, and 20.5% of unvaccinated patients with the original, Gamma, Delta, and Omicron variants, respectively, were admitted to the ICU; 20.3%, 34.5%, 21.6%, and 9.2%, respectively, required invasive ventilation; and 14.7%, 26.5%, 16.7%, and 2.1%, respectively, died. The mortality rate was considerably lower in unvaccinated pregnant patients, except for patients with the Omicron variant.

Studies including the general population have reported that different VOCs and vaccination statuses affect the clinical symptoms and outcomes of patients with COVID-19 [41,42]. A review concluded that, compared with the original variant, the Alpha, Beta, and Gamma variants were associated with higher risks of hospitalization and ICU admission, with the Beta variant having the highest risk of ICU admission [43]. A meta-analysis reported higher risks of hospitalization, ICU admission, and mortality in patients infected with the Beta and Delta variants than in those infected with the Alpha and Gamma variants. The meta-analysis also concluded that, compared with the original SARS-CoV-2 variant, all SARS-CoV-2 VOCs are associated with a higher risk of severe outcomes [44].

In this study, postpartum patients were less likely to have comorbidities, although the comorbidities varied between the VOC subgroups. Patients in the puerperal group also presented with different clinical findings based on the VOC and their vaccination status. Patients in the puerperal group who were unvaccinated were more likely to have an SpO_2_ < 95% on room air when infected with the original variant. Unvaccinated patients in the puerperal group with the original, Gamma, and Omicron variants were more likely to be admitted to the ICU and to require invasive ventilatory support, and those with the original and Gamma variants were more likely to die.

Among the vaccinated puerperal patients, those in the Omicron variant subgroup had a higher risk of having an SpO_2_ < 95% on room air. Vaccinated puerperal patients in the Omicron subgroup also had a higher chance of ICU admission and requiring invasive ventilatory support. Those in the Gamma subgroup had an increased risk of death. These findings suggest that puerperal women have a higher risk of severe outcomes than pregnant women, especially when unvaccinated.

Previous observational studies and surveillance data regarding vaccination during pregnancy are reassuring. Most patients in the previous studies were administered the Pfizer/BioNTech or Moderna vaccines, and no adverse outcomes or side effects of vaccination during pregnancy were reported [45,46,47,48]. In Brazil, pregnant women are administered the Sinovac/Butantan and Pfizer/Wyeth vaccines [49]. Among the hospitalized pregnant and postpartum patients with severe COVID-19, those who received two doses of a COVID-19 vaccine had a 46% reduction in the odds of ICU admission, an 81% reduction in the odds of requiring invasive ventilatory support, and an 80% reduction in the odds of death, compared with those who did not receive any COVID-19 vaccination [20]. Among the general population, several studies have indicated that COVID-19 vaccination protects against severe outcomes caused by SARS-CoV-2 variants, including the Omicron variant [50,51,52]. The results of the current study support these previous findings and suggest that vaccines effectively reduce adverse outcomes, including ICU admission, the requirement for invasive ventilatory support, and death in pregnant and puerperal patients. Therefore, vaccination is an important public health strategy globally, and pregnant and postpartum women should be included in randomized clinical trials [19,20,53].

This study has several strengths, including using a large dataset with nationwide coverage and no duplicates that included hospitalized obstetric patients with COVID-19 confirmed via RT-PCR, VOC data, and vaccination status. Similar studies neither reported these data nor divided the obstetric population into pregnant and puerperal groups.

However, this study has limitations, including its retrospective nature based on a secondary database analysis. Although reporting COVID-19-related hospital admissions is compulsory in Brazil, it is not guaranteed that all patients with COVID-19 who were hospitalized were included in the database. Bias due to missing or inaccurate data cannot be ruled out. Although we have information on vaccination against COVID-19 in SIVEP-Gripe, the field “type of vaccine administered” is incomplete and has several missing entries and, hence, was not included in this analysis. Asymptomatic or mildly symptomatic patients are not reported in the SIVEP-Gripe; therefore, an analysis of the non-hospitalized population was not possible. In addition, the database used in this study does not include information regarding other obstetric variables such as gestational age, delivery mode, date of birth, comorbidities, week of pregnancy at diagnosis, and perinatal data. As it is an anonymous database, the data cannot be linked with the public database of birth registers and mode of delivery.

As postpartum women have higher risks of developing severe forms of COVID-19, especially when unvaccinated, health care strategies are necessary to implement vaccination and manage postpartum patients with COVID-19. The risk of severe COVID-19 in puerperal patients should not be underestimated, and health care workers should closely monitor all postpartum women infected with SARS-CoV-2, especially those not vaccinated. The vaccination of pregnant and postpartum patients should become a public health priority.

## 5. Conclusions

In this retrospective cross-sectional study, we analyzed the impact of the original, Gamma, Delta, and Omicron variants of SARS-CoV-2 in vaccinated and unvaccinated pregnant and postpartum women from February 2020 to July 2022. In comparison with unvaccinated pregnant women, those who were unvaccinated during the postpartum period had a higher chance of requiring invasive ventilatory support and ICU admission when infected with the original, Gamma, and Omicron variants and a higher chance of mortality when infected with the original, and Gamma variants. Among those vaccinated, postpartum women had a higher chance of requiring invasive ventilatory support and ICU admission when infected with the Omicron variant, and a higher chance of mortality when infected with the Gamma variant.

Our findings suggest that vaccines effectively reduce adverse outcomes, including ICU admission, the requirement for invasive ventilatory support, and death in both pregnant and puerperal patients. This study is unique in that patients were divided into pregnant and puerperal groups, as well as virus variant subgroups and vaccination status subgroups, for a more accurate analysis of the obstetric population. We hope that this research increases the awareness of the disease severity of SARS-CoV-2 VOCs in this specific population, particularly in those who are unvaccinated, and makes the public aware of the importance and priority of vaccination.

## Figures and Tables

**Figure 1 vaccines-10-02172-f001:**
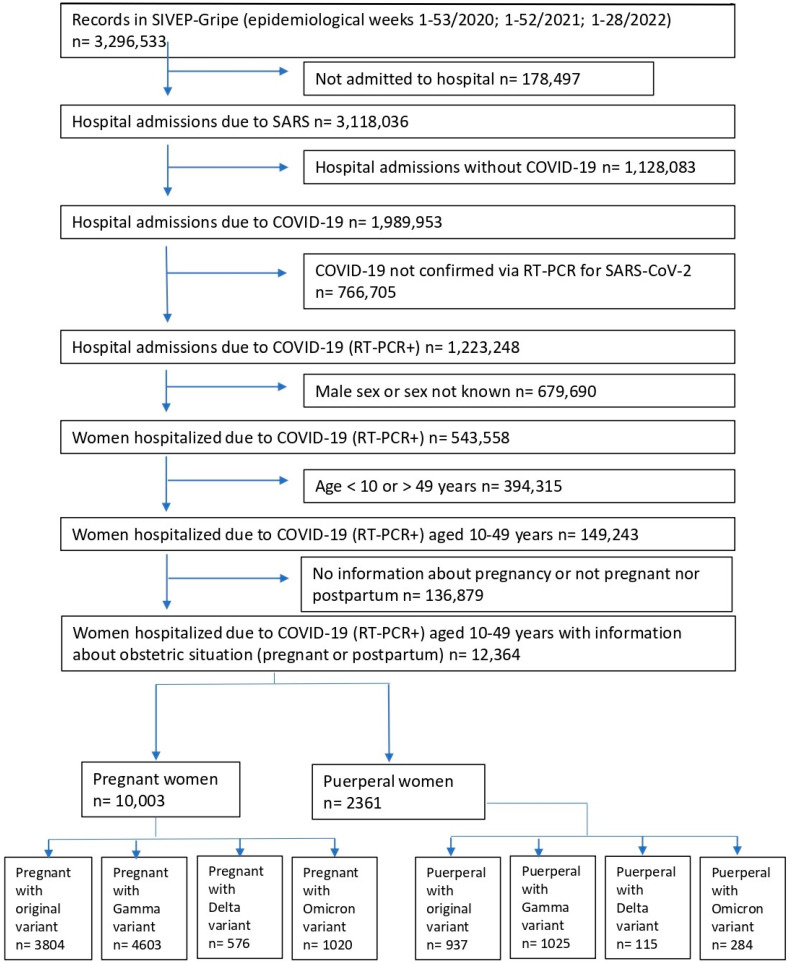
Flow chart of overall hospital admissions showing final counts of pregnant and puerperal women admitted with their categorization based on infection by specific coronavirus variants.

**Table 1 vaccines-10-02172-t001:** Patient characteristics.

Variant	Age (Years)	Pregnant	Puerperal	*p*
Original (*n* = 4741)	<20	261 (6.9%)	67 (7.2%)	0.2567 ^C^
	20–34	2526 (66.4%)	596 (63.6%)	
	≥35	1017 (26.7%)	274 (29.2%)	
Gamma (*n* = 5628)	<20	243 (5.3%)	62 (6.0%)	0.5902 ^C^
	20–34	2976 (64.7%)	662 (64.6%)	
	≥35	1384 (30.1%)	301 (29.4%)	
Delta (*n* = 691)	<20	39 (6.8%)	11 (9.6%)	0.5666 ^C^
	20–34	391 (67.9%)	75 (65.2%)	
	≥35	146 (25.3%)	29 (25.2%)	
Omicron (*n* = 1304)	<20	85 (8.3%)	30 (10.6%)	0.3260 ^C^
	20–34	722 (70.8%)	189 (66.5%)	
	≥35	213 (20.9%)	65 (22.9%)	
**Variant**	**Skin color/ethnicity**	**Pregnant**	**Puerperal**	** *p* **
Original (*n* = 3842)	White	1301 (42.1%)	289 (38.6%)	0.07546 ^F^
	Black	210 (6.8%)	52 (6.9%)	
	Yellow	38 (1.2%)	3 (0.4%)	
	Brown	1531 (49.5%)	401 (53.5%)	
	Indigenous	13 (0.4%)	4 (0.5%)	
Gamma (*n* = 4812)	White	1917 (48.6%)	407 (46.8%)	0.5847 ^F^
	Black	230 (5.8%)	56 (6.4%)	
	Yellow	32 (0.8%)	6 (0.7%)	
	Brown	1754 (44.5%)	397 (45.6%)	
	Indigenous	9 (0.2%)	4 (0.5%)	
Delta (*n* = 594)	White	242 (48.8%)	50 (51.0%)	0.9395 ^F^
	Black	36 (7.3%)	5 (5.1%)	
	Yellow	3 (0.6%)	0 (0.0%)	
	Brown	211 (42.5%)	43 (43.9%)	
	Indigenous	4 (0.8%)	0 (0.0%)	
Omicron (*n* = 1176)	White	610 (65.9%)	143 (57.0%)	0.0039 ^F^
	Black	50 (5.4%)	10 (4.0%)	
	Yellow	9 (1.0%)	0 (0.0%)	
	Brown	255 (27.6%)	97 (38.6%)	
	Indigenous	1 (0.1%)	1 (0.4%)	
**Variant**	**Education level**	**Pregnant**	**Puerperal**	** *p* **
Original (*n* = 2002)	No schooling	7 (0.4%)	3 (0.8%)	0.5112 ^F^
	Up to high school	394 (24.1%)	84 (22.7%)	
	High school	898 (55.1%)	200 (53.9%)	
	College	332 (20.4%)	84 (22.6%)	
Gamma (*n* = 2365)	No schooling	15 (0.8%)	6 (1.4%)	0.4348 ^F^
	Up to high school	469 (24.3%)	112 (25.8%)	
	High school	1051 (54.4%)	236 (54.4%)	
	College	396 (20.5%)	80 (18.4%)	
Delta (*n* = 287)	No schooling	0 (0.0%)	0 (0.0%)	0.5912 ^F^
	Up to high school	72 (30.1%)	11 (22.9%)	
	High school	112 (46.9%)	24 (50.0%)	
	College	55 (23.0%)	13 (27.1%)	
Omicron (*n* = 586)	No schooling	2 (0.4%)	2 (1.9%)	0.3158 ^F^
	Up to high school	135 (28.2%)	31 (28.9%)	
	High school	249 (52.0%)	51 (47.7%)	
	College	93 (19.4%)	23 (21.5%)	
**Comorbidities**	**Variant**	**Pregnant**	**Puerperal**	** *p* **
Chronic cardiovascular disease	Original (*n* = 1889)	239/1306 (18.3%)	89/583 (15.3%)	0.1230 ^C^
	Gamma (*n* = 2084)	245/1409 (17.4%)	84/675 (12.4%)	0.0046 ^C^
	Delta (*n* = 256)	29/179 (16.2%)	8/77 (10.4%)	0.3082 ^C^
	Omicron (*n* = 441)	49/265 (18.5%)	13/176 (7.4%)	0.0017 ^C^
Asthma	Original (*n* = 1847)	160/1288 (12.4%)	34/559 (6.1%)	0.0001 ^C^
	Gamma (*n* = 2040)	148/1373 (10.8%)	37/667 (5.5%)	0.0002 ^C^
	Delta (*n* = 256)	23/181 (12.7%)	6/75 (8.0%)	0.3871 ^C^
	Omicron (*n* = 429)	26/257 (10.1%)	6/172 (3.5%)	0.0176 ^C^
Diabetes	Original (*n* = 1887)	274/1319 (20.8%)	65/568 (11.4%)	<0.0001 ^C^
	Gamma (*n* = 2134)	337/1448 (23.3%)	77/686 (11.2%)	<0.0001 ^C^
	Delta (*n* = 267)	48/188 (25.5%)	8/79 (10.1%)	0.0079 ^C^
	Omicron (*n* = 446)	62/268 (23.1%)	10/178 (5.6%)	<0.0001 ^C^
Obesity	Original (*n* = 1822)	181/1263 (14.3%)	64/559 (11.4%)	0.1122 ^C^
	Gamma (*n* = 2142)	368/1451 (25.4%)	111/691 (16.1%)	<0.0001 ^C^
	Delta (*n* = 259)	39/183 (21.3%)	8/76 (10.5%)	0.0610 ^C^
	Omicron (*n* = 425)	32/249 (12.9%)	10/176 (5.7%)	0.0229 ^C^

^C^ Chi-square test was used to compare the groups, ^F^ Fisher’s exact test was used to compare the groups.

**Table 2 vaccines-10-02172-t002:** Signs and symptoms of COVID-19 in pregnant and postpartum women.

Signs and Symptoms	Variant	Unvaccinated			Vaccinated (at Least One Dose)		
		Pregnant	Puerperal	Puerperal vs. Pregnant OR (95% CI)	Pregnant	Puerperal	Puerperal vs. Pregnant OR (95% CI)
Fever	Original (*n* = 4150)	2283/3345 (68.3%)	500/805 (62.1%)	0.76 (0.65–0.90)	-	-	-
	Gamma (*n* = 2839)	1283/1995 (64.3%)	241/459 (52.5%)	0.61 (0.50–0.76)	204/321 (63.6%)	32/64 (50.0%)	0.57 (0.32–1.02)
	Delta (*n* = 529)	131/206 (63.6%)	30/47 (63.8%)	1.01 (0.50–2.09)	146/237 (61.6%)	20/39 (51.3%)	0.66 (0.31–1.38)
	Omicron (*n* = 881)	108/190 (56.8%)	28/66 (42.4%)	0.56 (0.30–1.02)	244/491 (49.7%)	40/134 (29.9%)	0.43 (0.28–0.66)
Cough	Original (*n* = 4272)	2674/3454 (77.4%)	576/818 (70.4%)	0.69 (0.58–0.83)	-	-	-
	Gamma (*n* = 2952)	1689/2079 (81.2%)	335/473 (70.8%)	0.56 (0.44–0.71)	273/331 (82.5%)	51/69 (73.9%)	0.60 (0.32–1.17)
	Delta (*n* = 568)	175/223 (78.5%)	40/52 (76.9%)	0.91 (0.43–2.06)	194/252 (77.0%)	28/41 (68.3%)	0.64 (0.30–1.45)
	Omicron (*n* = 953)	140/205 (68.3%)	43/68 (63.2%)	0.80 (0.43–1.49)	359/529 (67.9%)	96/151 (63.6%)	0.83 (0.56–1.23)
Sore throat	Original (*n* = 3656)	825/2960 (27.9%)	182/696 (26.1%)	0.92 (0.76–1.11)	-	-	-
	Gamma (*n* = 2567)	488/1810 (27.0%)	96/414 (23.2%)	0.82 (0.63–1.06)	71/281 (25.3%)	13/62 (21.0%)	0.79 (0.37–1.58)
	Delta (*n* = 486)	48/186 (25.8%)	8/45 (17.8%)	0.62 (0.23–1.48)	69/222 (31.1%)	9/33 (27.3%)	0.83 (0.32–1.97)
	Omicron (*n* = 833)	70/182 (38.5%)	15/58 (25.9%)	0.56 (0.27–1.12)	177/462 (38.3%)	40/131 (30.5%)	0.71 (0.45–1.09)
Dyspnea	Original (*n* = 4086)	2024/3305 (61.2%)	467/781 (59.8%)	0.94 (0.80–1.11)	-	-	-
	Gamma (*n* = 2913)	1493/2051 (72.8%)	314/470 (66.8%)	0.75 (0.60–0.94)	231/326 (70.9%)	44/66 (66.7%)	0.82 (0.45–1.52)
	Delta (*n* = 545)	149/216 (69.0%)	36/51 (70.6%)	1.08 (0.53–2.27)	147/238 (61.8%)	20/40 (50.0%)	0.62 (0.29–1.29)
	Omicron (*n* = 818)	58/174 (33.3%)	16/61 (26.2%)	0.71 (0.35–1.42)	117/443 (26.4%)	52/140 (37.1%)	1.64 (1.07–2.50)
Respiratory discomfort	Original (*n* = 3882)	1565/3123 (50.1%)	413/759 (54.4%)	1.19 (1.01–1.39)	-	-	-
	Gamma (*n* = 2762)	1143/1944 (58.8%)	267/448 (59.6%)	1.03 (0.83–1.28)	168/303 (55.4%)	35/67 (52.2%)	0.88 (0.50–1.55)
	Delta (*n* = 510)	113/203 (55.7%)	22/47 (46.8%)	0.70 (0.35–1.39)	105/222 (47.3%)	11/38 (28.9%)	0.46 (0.19–1.00)
	Omicron (*n* = 807)	42/170 (24.7%)	14/60 (23.3%)	0.92 (0.43–1.92)	127/438 (29.0%)	42/139 (30.2%)	1.06 (0.68–1.63)
SpO_2_ < 95%	Original (*n* = 3780)	1020/3043 (33.5%)	340/737 (46.1%)	1.70 (1.44–2.00)	-	-	-
	Gamma (*n* = 2780)	1120/1964 (57.0%)	264/451 (58.5%)	1.06 (0.86–1.32)	163/299 (54.5%)	34/66 (51.5%)	0.89 (0.50–1.57)
	Delta (*n* = 517)	112/205 (54.6%)	32/47 (68.1%)	1.77 (0.86–3.74)	93/227 (41.0%)	17/38 (44.7%)	1.16 (0.55–2.46)
	Omicron (*n* = 783)	31/170 (18.2%)	12/57 (21.1%)	1.19 (0.51–2.63)	67/426 (15.7%)	35/130 (26.9%)	1.97 (1.20–3.22)
Diarrhea	Original (*n* = 3539)	404/2879 (14.0%)	74/660 (11.2%)	0.77 (0.58–1.01)	-	-	-
	Gamma (*n* = 2501)	220/1761 (12.5%)	37/406 (9.1%)	0.70 (0.47–1.02)	25/271 (9.2%)	7/63 (11.1%)	1.23 (0.43–3.11)
	Delta (*n* = 472)	17/183 (9.3%)	5/44 (11.4%)	1.25 (0.34–3.82)	22/212 (10.4%)	1/33 (3.0%)	0.27 (0.01–1.80)
	Omicron (*n* = 583)	13/167 (7.8%)	2/56 (3.6%)	0.44 (0.05–2.04)	22/416 (5.3%)	7/122 (5.7%)	1.09 (0.38–2.73)
Vomiting	Original (*n* = 3539)	399/2884 (13.8%)	46/655 (7.0%)	0.47 (0.33–0.65)	-	-	-
	Gamma (*n* = 2510)	249/1771 (14.1%)	30/407 (7.4%)	0.49 (0.32–0.73)	35/271 (12.9%)	7/61 (11.5%)	0.87 (0.31–2.14)
	Delta (*n* = 470)	21/182 (11.5%)	1/44 (2.3%)	0.18 (0.00–1.18)	24/211 (11.4%)	2/33 (6.1%)	0.50 (0.06–2.21)
	Omicron (*n* = 770)	24/170 (14.1%)	2/55 (3.6%)	0.23 (0.03–0.98)	54/423 (12.8%)	6/122 (4.9%)	0.35 (0.12–0.85)
Abdominal pain	Original (*n* = 1886)	159/1562 (10.2%)	26/324 (8.0%)	0.77 (0.48–1.20)	-	-	-
	Gamma (*n* = 2476)	177/1750 (10.1%)	35/401 (8.7%)	0.85 (0.56–1.25)	29/263 (11.0%)	3/62 (4.8%)	0.41 (0.08–1.40)
	Delta (*n* = 466)	23/181 (12.7%)	2/44 (4.5%)	0.33 (0.04–1.42)	21/208 (10.1%)	4/33 (12.1%)	1.23 (0.29–4.02)
	Omicron (*n* = 761)	18/168 (10.7%)	3/55 (5.5%)	0.48 (0.09–1.75)	56/415 (13.5%)	9/123 (7.3%)	0.51 (0.21–1.07)
Fatigue	Original (*n* = 1933)	394/1595 (24.7%)	71/338 (21.0%)	0.81 (0.60–1.08)	-	-	-
	Gamma (*n* = 2576)	640/1820 (35.2%)	127/411 (30.9%)	0.82 (0.65–1.04)	113/279 (40.5%)	21/66 (31.8%)	0.69 (0.37–1.25)
	Delta (*n* = 478)	52/187 (27.8%)	17/45 (37.8%)	1.57 (0.74–3.27)	62/213 (29.1%)	7/33 (21.2%)	0.66 (0.23–1.66)
	Omicron (*n* = 768)	38/168 (22.6%)	9/58 (15.5%)	0.63 (0.25–1.45)	75/417 (18.0%)	24/125 (19.2%)	1.08 (0.62–1.84)
Anosmia	Original (*n* = 1984)	442/1636 (27.0%)	78/348 (22.4%)	0.78 (0.58–1.03)	-	-	-
	Gamma (*n* = 2517)	373/1778 (21.0%)	55/401 (13.7%)	0.60 (0.43–0.82)	59/275 (21.5%)	10/63 (15.9%)	0.69 (0.30–1.48)
	Delta (*n* = 468)	37/180 (20.6%)	4/44 (9.1%)	0.39 (0.09–1.18)	44/210 (21.0%)	6/34 (17.6%)	0.81 (0.26–2.16)
	Omicron (*n* = 748)	7/164 (4.3%)	2/54 (3.7%)	0.86 (0.08–4.73)	14/407 (3.4%)	2/123 (1.6%)	0.46 (0.05–2.07)
Ageusia	Original (*n* = 1958)	400/1615 (24.8%)	66/343 (19.2%)	0.72 (0.53–0.97)	-	-	-
	Gamma (*n* = 2517)	332/1777 (18.7%)	55/403 (13.6%)	0.69 (0.50–0.94)	64/276 (23.2%)	8/61 (13.1%)	0.50 (0.20–1.13)
	Delta (*n* = 468)	37/178 (20.8%)	5/44 (11.4%)	0.49 (0.14–1.37)	42/211 (19.9%)	7/35 (20.0%)	1.01 (0.35–2.57)
	Omicron (*n* = 746)	7/163 (4.3%)	1/54 (1.9%)	0.42 (0.01–3.41)	18/406 (4.4%)	3/123 (2.4%)	0.54 (0.10–1.89)

Abbreviations: OR, odds ratio; 95% CI, 95% confidence interval; SpO_2_, oxygen saturation on room air.

**Table 3 vaccines-10-02172-t003:** Outcomes of pregnant and postpartum women with COVID-19.

Outcome	Variant	Unvaccinated			Vaccinated (with at Least One Dose)		
		Pregnant	Puerperal	Puerperal vs. Pregnant OR (95% CI)	Pregnant	Puerperal	Puerperal vs. Pregnant OR (95% CI)
ICU admission (*n* = 9202)	Original (*n* = 4436)	833/3566 (23.4%)	310/870 (35.6%)	1.82 (1.54–2.13)	-	-	-
	Gamma (*n* = 3051)	795/2157 (36.9%)	235/490 (48.0%)	1.58 (1.29–1.93)	91/334 (27.2%)	28/70 (40.0%)	1.78 (1.00–3.13)
	Delta (*n* = 579)	90/225 (40.0%)	26/51 (51.0%)	1.56 (0.81–3.01)	75/256 (29.3%)	16/47 (34.0%)	1.24 (0.60–2.51)
	Omicron (*n* = 1136)	23/257 (8.9%)	18/88 (20.5%)	2.61 (1.25–5.38)	59/623 (9.5%)	28/168 (16.7%)	1.91 (1.13–3.17)
Invasive respiratory support (*n* = 8904)	Original (*n* = 4237)	300/3389 (8.9%)	172/848 (20.3%)	2.62 (2.13–3.22)	-	-	-
	Gamma (*n* = 2964)	411/2105 (19.5%)	163/472 (34.5%)	2.17 (1.75–2.70)	39/319 (12.2%)	12/68 (17.6%)	1.54 (0.76–3.12)
	Delta (*n* = 576)	34/225 (15.1%)	11/51 (21.6%)	1.54 (0.72–3.30)	20/254 (7.9%)	4/46 (8.7%)	1.11 (0.36–3.42)
	Omicron (*n* = 1127)	8/252 (3.2%)	8/87 (9.2%)	3.09 (1.12–8.50)	16/619 (2.6%)	12/169 (7.1%)	2.88 (1.34–6.21)
Death (*n* = 9206)	Original (*n* = 4449)	231/3563 (6.5%)	130/886 (14.7%)	2.48 (1.96–3.13)	-	-	-
	Gamma (*n* = 3007)	325/2117 (15.4%)	128/483 (26.5%)	1.99 (1.56–2.53)	22/341 (6.5%)	10/66 (15.2%)	2.58 (1.03–6.06)
	Delta (*n* = 559)	23/222 (10.4%)	8/48 (16.7%)	1.73 (0.62–4.36)	7/243 (2.9%)	1/46 (2.2%)	0.75 (0.02–6.07)
	Omicron (*n* = 1191)	6/267 (2.2%)	2/94 (2.1%)	0.95 (0.09–5.41)	7/659 (1.1%)	6/171 (3.5%)	3.38 (0.93–11.93)

Abbreviations: OR, odds ratio; 95% CI, 95% confidence interval; ICU, intensive care unit.

## Data Availability

The data that support the findings of this study are available in GitHub repository at https://github.com/observatorioobstetrico/COVID-19-impact-of-original-Gamma-Delta-and-Omicron-variants-of-SARS-CoV-2-in-vaccinated-and-unv. (accessed on 23 October 2022). These data were derived from the following resources available in the public domain: https://opendatasus.saude.gov.br/dataset/srag-2020 and https://opendatasus.saude.gov.br/dataset/srag-2021-e-2022 (accessed on 25 July 2022).

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
