# Peer review of "COVID-19: Impact of Original, Gamma, Delta, and Omicron Variants of SARS-CoV-2 in Vaccinated and Unvaccinated Pregnant and Postpartum Women"

_vaccines, 2022, doi:10.3390/vaccines10122172_

Round 1

Reviewer 1 Report

The manuscript titled “COVID-19: impact of original, Gamma, Delta, and Omicron variants of SARS-CoV-2 in vaccinated and unvaccinated pregnant and postpartum women” by Serra et al is an interesting study. In this study authors compares impact of vaccination on vaccinated and unvaccinated pregnant and postpartum women. They have found that unvaccinated population are more vulnerable towards severe SARS-CoV-2 infection and even death. Also they have observed that postpartum women are more prone to severe SARS-CoV-2 infection. The manuscript is well drafted and well organized, and the study is also relevant to the field. However I found few shortcomings in the present form of the manuscript which I believe needs to be fixed before publication.

Major points:

1. I think two vaccines are majorly used in Brazil SinoVac and CoronaVac, I am just wondering if author could analyze which vaccine is more effective against pregnant and postpartum women in their analysis that would definitely add value in this study.

2. Secondly Omicron is having several subtypes including highly infectious BA2 variants , if author could also analyzed the impact of different Omicron subtype on pregnant and postpartum women during their study period it would also provide novel insight 

Minor points:

There are some grammatical mistakes, typos in the manuscript, before final submission authors should thoroughly checked the manuscript.

Author Response

Dear Reviewer, thank you for taking out the time to review our manuscript and for your valuable comments.

Please find below our point-by-point response for each of your comment.

  • I think two vaccines are majorly used in Brazil SinoVac and CoronaVac, I am just wondering if author could analyze whichvaccine is more effective against pregnant and postpartumwomen in their analysis that would definitely add value in thisstudy.

Response: The field about the manufacturer of the vaccine in the SIVEP-Gripe database is an open and sparsely populated field (a little more than 10% of entries have this data). Since this is an open field, even when there is information filled in this field, it may not be always reliable.

However, the information about the doses received by the pregnant and postpartum population in Brazil is available via the National Immunization Program database. Information regarding the number of doses of vaccine obtained from the dashboard created by the Brazilian Obstetric Observatory (https://observatorioobstetrico.shinyapps.io/vacinacao-covid19/ accessed at Dec 7, 2022), is presented below:

Vaccine used

1st dose

2nd dose or single dose

n

%

N

%

AstraZeneca

60980

5,91%

39335

4,19%

CoronaVac

302177

29,30%

282354

30,09%

Janssen

39

0,00%

32

0,00%

Pfizer

668021

64,78%

616751

65,72%

As this information is available on another anonymized database, it is not possible to study which vaccine is more effective against pregnant and postpartum women.

We included this information in the Discussion section of the revised manuscript as below:

“Although we have information on vaccination against COVID-19 in SIVEP-Gripe, the field “type of vaccine administered” is incomplete and has several missing entries and, hence, was not included in this analysis.”

  • Secondly Omicron is having several subtypes including highly infectious BA2 variants, if author could also analyze the impactof different Omicron subtype on pregnant and postpartumwomen during their study period it would also provide novelinsight.

Response: Thank you for your suggestion. We decided do not divide Omicron Group based on the different subtypes because this group was not as big as the others. If we had split Omicron in four subtypes (the ones that were prevalent during that time), the analysis of the outcomes would be impaired. Please check their occurrence below:

  • There are some grammatical mistakes, typos in the manuscript,

before final submission authors should thoroughly checked the

manuscript.

Response: Thank you for pointing this out. We had the manuscript proofread for English Language by professional service providers. The revisions can be seen as track changes in the revised manuscript.

Reviewer 2 Report

Serra et al, summarize in a retrospective study of pregnant and postpartum women infected with four variants of SARS-CoV-2 in Brazil to assess differences in severity of disease with and without prior vaccination. Those women without prior vaccination had worse outcomes and in general postpartum women had worse outcomes.  The authors conclude that vaccination is a priority for these groups and that infected, postpartum women may need additional attention.

Possible improvements:

I know the focus of this project was pregnant and puerperal women and other studies have already compared these women to a non-pregnant and non-puerperal group of women, but I think if a similar group had been used here, the impact of this data would have been so much greater. With that data available to you, why would you not include it?

It would have been good to see statistics of vaccination status to justify splitting the groups into vaccinated and unvaccinated.

Author Response

Dear Reviewer, we thank you for having reviewed our manuscript and giving your valuable comments.

Please find below point-by-point responses for each of your comment.

  • English language and style are fine/minor spell check required

Response: Thank you for pointing this out. We sent the manuscript for English Language check. The revisions can be seen as track changes in the revised manuscript.

  • I know the focus of this project was pregnant and puerperal women and other studies have already compared these women to a non-pregnant and non-puerperal group of women, but I think if a similar group had been used here, the impact of this data would have been so much greater. With that data available to you, why would you not include it?

Response: We performed a retrospective analysis of the records of the same database (SIVEP-Gripe) that included the data of female patients aged 10 to 49 years hospitalized because of severe COVID-19 disease (had positive RT-PCR for SARS-CoV-2), from February 17, 2020, to January 02, 2021. They were separated into 3 groups: pregnant, puerperal, and neither pregnant nor puerperal. After general comparisons, adjustments for confounding variables (propensity score matching [PSM]) were made using demographic and clinical characteristics, disease progression (admission to the intensive care unit [ICU] and invasive or noninvasive ventilatory support), and outcome (cure or death). After PSM, prognosis of puerperal women was worse than that for pregnant women with respect to admission to the ICU, invasive ventilatory support, and death, with OR (95% CI) 1.97 (1.55 – 2.50), 2.71 (1.78 – 4.13), and 2.51 (1.79 – 3.52), respectively. Outcomes for postpartum women was similar to those for non-pregnant and non-puerperal group of women (Table 3).

This is an open access article and can be accessed on: https://dx.plos.org/10.1371/journal.pone.0259911

The main finding of the article cited above is the worse prognosis of puerperal women compared with pregnant women. These results motivated us to assess whether the results were consistent across the different variants and, therefore, we only compared pregnant and postpartum women in the current study.

We included this data in the Introduction section to clarify the reason for making this.

  • It would have been good to see statistics of vaccination status to justify splitting the groups into vaccinated and unvaccinated.

Response: The information we have about vaccination status does not represent the real percentage of vaccinated pregnant or postpartum women as these numbers and their condition are very dynamic. It is a cumulative number that changes dynamically with time. It is difficult to estimate the number of pregnant women and their exact vaccination status. However, it is possible to check the total number of vaccinated pregnant and postpartum women in the population.

According to the dashboard created by the Brazilian Obstetric Observatory (https://observatorioobstetrico.shinyapps.io/vacinacao-covid19/ accessed at Dec 7, 2022), 1,031,217 pregnant and postpartum women received the first dose, and 940,475 received the second or the single dose in Brazil.

Round 2

Reviewer 1 Report

I am satisfied with the answers and current form of the manuscript, please accept it for publication.